# Assessment for the Sustainable Development of Components of the Tourism and Recreational Potential of Rural Areas of the Aktobe Oblast of the Republic of Kazakhstan

Kuat Saparov [1], Miroslava Omirzakova [1,\*], Aigul Yeginbayeva [1], Aigul Sergeyeva [2,\*], Kairat Saginov [1] and Gulnash Askarova [2]

[1] Department of Physical and Economical Geography, L.N. Gumilyov Eurasian National University, Astana 010000, Kazakhstan; saparov_kt@enu.kz (K.S.); yeginbayeva_aye@enu.kz (A.Y.); saginov_km@enu.kz (K.S.)
[2] Department of Geography and Tourism, K. Zhubanov Aktobe Regional University, Aktobe 030000, Kazakhstan; gaskarova@zhubanov.edu.kz
\* Correspondence: omirzakova_mzh@enu.kz (M.O.); asergeyeva@zhubanov.edu.kz (A.S.)

**Abstract:** The assessment of sustainable tourism development in the rural areas of the Aktobe oblast of Kazakhstan involved thoroughly analyzing multiple dimensions. Environmental, socio-economic, and cultural sustainability aspects were considered to comprehensively understand the region's tourism potential. The study began by evaluating the available tourism resources in rural Aktobe. This included assessing natural attractions such as landscapes, wildlife, and geological features, as well as cultural heritage sites and infrastructure like accommodation facilities and transportation networks. A crucial aspect of the study was to analyze the environmental impact of tourism activities in rural areas. This involved evaluating the effects on ecosystems and natural resources. The measures for conserving these resources were also identified. Another focus was on the socio-cultural aspects of tourism development. The study aimed to preserve local traditions, cultural heritage, and community identity amidst tourism growth. Strategies for achieving socio-cultural sustainability were devised. Ranking methods were employed to identify key factors influencing rural tourism development. These methods helped prioritize areas for improvement and resource allocation. A balanced approach was adopted to assess the interaction between different dimensions of sustainability. This ensured that environmental, economic, and socio-cultural aspects were considered equally to achieve overall sustainable tourism development. ArcGIS 10 was used for data analysis and visualization. Maps and charts were created to represent spatial and statistical information, aiding in identifying trends and patterns. The study findings were crucial for identifying priority areas for infrastructure development and formulating strategies and programs for rural tourism promotion. The study aimed to ensure that tourism development aligns with the principles of sustainable development, benefiting both the local communities and the environment. The study provided valuable insights into the current status of rural tourism in Aktobe oblast and offered recommendations for sustainable development, contributing to the region's long-term prosperity.

**Keywords:** rural tourism; resource potential; balance method; geographic information system; buffer analysis; Aktobe oblast

## 1. Introduction

Rural development is of strategic importance to the country as it significantly impacts food security, which is largely dependent on good governance, institutional capacity, and policy and implementation. Any territory represents a living, dynamic reality by its nature, and its quality and individuality are determined by directly or indirectly measurable elements of natural conditions, infrastructure and principles of economic activity. The

originality of the territory is reflected in its economic and social features, demographic and cultural aspects, natural-ecological and political characteristics [1–3].

The comprehensive development of rural tourism is one of the urgent problems of Kazakhstan. Rural tourism development models based on sustainable development can develop when they meet the needs of local residents and respond to the needs of tourists [4,5]. The development of tourism activities in rural areas can create positive effects such as improving the quality of life of the local population, creating jobs, preserving cultural heritage, developing business networks, and shaping the image of the oblast [6–8].

Rural tourism is part of a variety of tourism types, such as ecotourism, farm tourism, adventure tourism, gastronomic tourism, and cultural tourism, forming a complex and multifaceted sector with an ever-expanding variety of opportunities. Rural tourism covers three key aspects: the material characteristics of the countryside, the interaction of tourists with these characteristics, and the cultural or symbolic value of the area. It is a form of tourism located in rural areas that brings together a variety of activities and services to develop and revitalize areas.

The development of rural tourism in each country depends on its unique natural and resource potential, as well as on the historical heritage, cultural attractions, gastronomic features, and natural attractions. Rural tourism is currently expanding rapidly in various countries around the world. In addition to well-known destinations such as Italy, Germany, France, China, Poland, and the Baltic countries, the best practices and experiences in rural tourism are being actively explored. Rural tourism has similarities with other types of tourism, and many of them can be successfully implemented in rural areas. Rural tourism can provide a variety of activities and experiences similar to spiritual, cultural, and adventure tourism, including forms such as farm tourism. However, it is essential to consider that only a few attractions can dilute the uniqueness of the experience. The process of redefining rural tourism at the national level, as it has been carried out for example in Malaysia and Portugal, shows that developing a new definition in a specific context can help better convey the experience of rural tourism destinations. This allows us to identify the unique characteristics of each destination and emphasize them as part of tourism promotion. As a result, although achieving a global consensus on the definition of rural tourism may be difficult, a definition specific to each country or region may be more realistic and useful [9].

Rural tourism in Kazakhstan is a promising direction that attracts the attention of public and private structures within the framework of a tourism development policy. Here is an overview of the current state of rural tourism in Kazakhstan, considering the current tourism development policy.

In recent years, several rural tourism development programs have been launched in Kazakhstan, such as "Baitak Zher", "On your own through Semirechye", etc. These programs aim to create new tourist routes, develop hotel and restaurant businesses, and train local residents in the field of tourism and other activities. The "Baytak Zher" project, launched in the fall of 2022, is being successfully implemented in three districts of the Akmola oblast: Zerenda, Sandyktau, and Korgalzhyn. Under the leadership of this project, 20 entrepreneurs completed a training incubation program covering the basics of eco- and ethnotourism, financial and project planning, and methods of promoting their brands.

While rural tourism presents a promising industry, it also brings forth serious challenges:

1. The need for more infrastructure. Rural areas often lack sufficient tourism infrastructure such as hotels, road networks, public transport, and other amenities, making it difficult to attract tourists;
2. A lack of funding. Investments in the development of rural tourism are limited due to a lack of financial resources, both from the state and from private investors;
3. Low awareness and marketing. Many rural areas face a lack of awareness of the potential of rural tourism and lack effective marketing strategies to attract tourists;

4. A lack of qualified personnel. In rural areas, there are not enough qualified tourism specialists, which makes it challenging to provide quality services and manage tourism enterprises;

5. The problems of preserving the natural environment. Intensive tourism hurts the natural environment of rural areas if appropriate measures are not taken to protect the environment and sustainably use resources;

6. Sociocultural aspects. Some rural communities will face challenges in adapting to the arrival of tourists, including changing lifestyles, preserving traditions, and maintaining privacy.

Solving these problems requires joint efforts on the part of the state, local authorities, entrepreneurs and communities to develop and implement rural tourism development strategies that take into account the needs of all stakeholders and ensure sustainable development.

*Literature Review*

Several researchers have proposed new models of rural tourism at different stages and concluded that innovation can be promoted by meeting the needs of tourists and providing appropriate products [10–12]. Cawley and Gillmor define rural tourism in terms of resource potential. Rural tourism is a new socio-economic phenomenon focused on using the resource potential of rural areas to create and promote tourism products to a wide range of people [13]. At the same time, it is advisable to understand the resource potential of the analyzed type of tourism as a set of interconnected and closely interacting potentials that are widely used in tourism activities, as well as new ones created in the process of carrying out such activities and using the factors of production of rural areas. Thus, the following components can be included in the system of resource potential of rural tourism: organizational potential; economic potential; information potential; historical and cultural potential; recreational potential; environmental potential, etc. [14–16].

Thanks to the ecological and tourism potential, which is part of the natural resource potential, the territories fulfill their socio-cultural load, contributing to sustainable development. The need to study the problem of formation, development, and assessment of the ecological and tourism potential of the region is determined not only by the need to diversify the use of resource opportunities but, to a greater extent, by the turbulent state of the social and labor sphere. In addition, the lack of general methodological approaches to assessing the ecological and tourism potential does not contribute to its effective use and, as a consequence, to the stable socio-economic development of territories [17]. The study of the natural prerequisites for the development of the tourism sector is traditionally the first stage in assessing the territory since natural resources are one of the factors that predetermine its use [18].

Of particular scientific interest is the study of N.A. Mozgunov, Trukhachev, Semiglazov, and Zyryanov, which in analyzing the existing specialized literature, identified the key factors in developing rural tourism in modern conditions. These include socioeconomic (the situation around regions with a high solvent demand for rural tourism services); settlement (the general condition of the rural population); psychological (the internal readiness of areas living in a particular territory to accept potential clients and the readiness of individuals to go on vacation to the countryside); natural (reasonably comfortable natural and climatic conditions); cultural and historical (the specific features of a particular area, places of residence and activities of famous people, etc.); infrastructure (transport accessibility, accommodation, food and related services); institutional (the interest of regional authorities in the development of rural tourism, development and implementation of development programs) [19–21].

It is crucial to emphasize that rural tourism presents a promising opportunity for the growth of small and medium-sized businesses, which are uniquely characterized by their focus on environmental preservation and social responsibility [22–24]. The environmental aspect of rural tourism is gaining significant importance in its organization. The actions of entrepreneurs and their competitors are not only evaluated based on their efforts to prevent

the use of harmful production methods and materials, but also from a social perspective. An entrepreneur who consistently implements an environmental protection program will, over time, build a positive reputation in society (region or industry). This, in turn, enhances its competitiveness and becomes a crucial competitive advantage that attracts a growing number of consumers [25–27].

In addition, doing business in the field of rural tourism is directly related to preserving the environment. This is objective because an ecologically clean area, untouched nature, the presence of recreational resources such as forests and rivers, and cultural and historical, architectural, and archaeological monuments are key conditions for the successful development of rural tourism in the region in the long term [28–31]. In this regard, the active participation of an entrepreneur in environmental protection activities and the restoration of cultural and historical monuments will become one of the indispensable conditions for the formation of constant tourist flows, and, consequently, increasing the income of individual citizens, regions and the state as a whole. This fact means that the entrepreneur is directly interested in protecting and reproducing existing natural and cultural resources [32–34]. Another conceptual feature of small business rural tourism is its pronounced social orientation. This is manifested in the intensification of the comprehensive development of human capital, that is, in rural areas the conditions necessary for proper recreation are created [35,36].

The countryside, with its attractive appearance and opportunity for city residents, satisfies their psychological needs and becomes a valuable resource in modern society. The commodification of this space, influenced by consumer culture, increases its value as landscape, environment, and products. However, this process can also lead to the abstraction and symbolization of rural space. Problems associated with excessive commodification are particularly felt in rural areas near large cities, reflecting a shift from overt to hidden forms of loss of rural identity [37].

Challenges relate to the unavailability of resources such as quality labor and investment, and the inability to use local resources to develop rural tourism. The lack of planning and government support demonstrates insufficient attention to the resource approach in research related to rural tourism [38].

More in-depth research can be undertaken using a resource-based approach to understand rural destinations' dynamic capabilities and identify how internal and external resources can be effectively identified, mobilized, harnessed and maintained. This will maximize rural tourism's benefits.

Considering the above, it can be argued that in modern conditions, rural tourism can become a significant source of additional income and sometimes the main income for the rural population. This is especially relevant for depressed regions characterized by deficient socioeconomic development [39].

Modern changes in the preferences of consumers of recreational services, as well as the growth of the environmental consciousness of vacationers, create a demand for new forms of tourism products that use alternative recreational resources, including the resources of rural areas. In Kazakhstan, where agriculture is a diversified production, there is a potential for the development of rural tourism as an additional factor in stimulating tourism in general. The culture, traditions, and hospitality of the Kazakh people can give a new impetus to the development of tourism in rural areas, in accordance with the principles of sustainable development [40–43].

The purpose of the study is to assess the sustainable development of components of the tourism and recreational potential of rural areas of the Aktobe oblast of the Republic of Kazakhstan.

Aktobe oblast is one of the regions of Kazakhstan where rural tourism can become a priority sector of the economy. However, the region's territory is characterized by insufficient recreational and geographical knowledge, which is one of the main factors hindering the development of the tourism industry, which is at the initial stage of formation. In this regard, the need arose for a comprehensive assessment of the environmental and

tourism potential of the Aktobe region and the determination of its regional specifics and prospects for use.

## 2. Study Area

Aktobe oblast is located in the western part of Kazakhstan. The length of the territory from west to east is about 800 km and from the north to the south about 700 km; the area is 300,629 km$^2$, with a population of 924,845 people (2022) [44]. The Aktobe oblast occupies an intermediate position between the Caspian lowland in the west, the Ustyurt plateau in the south, the Turan lowland in the southeast and the southern slopes of the Ural Mountains in the north. Its territory is mainly represented by plains intersected by river valleys that rise to a height of 100 to 200 m. Mugodzhary extends in the central part of the region. In the west of the region there is the Podural Plateau, which in the southwest turns into the Caspian Lowland. In the southeast of the region there are massifs of hilly sands—the Aral Karakum and Big and Small Badgers. In the northeast of the region stretches the Turgai plateau, riddled with ravines. The geographical location of Aktobe oblast in Kazakhstan is shown in Figure 1.

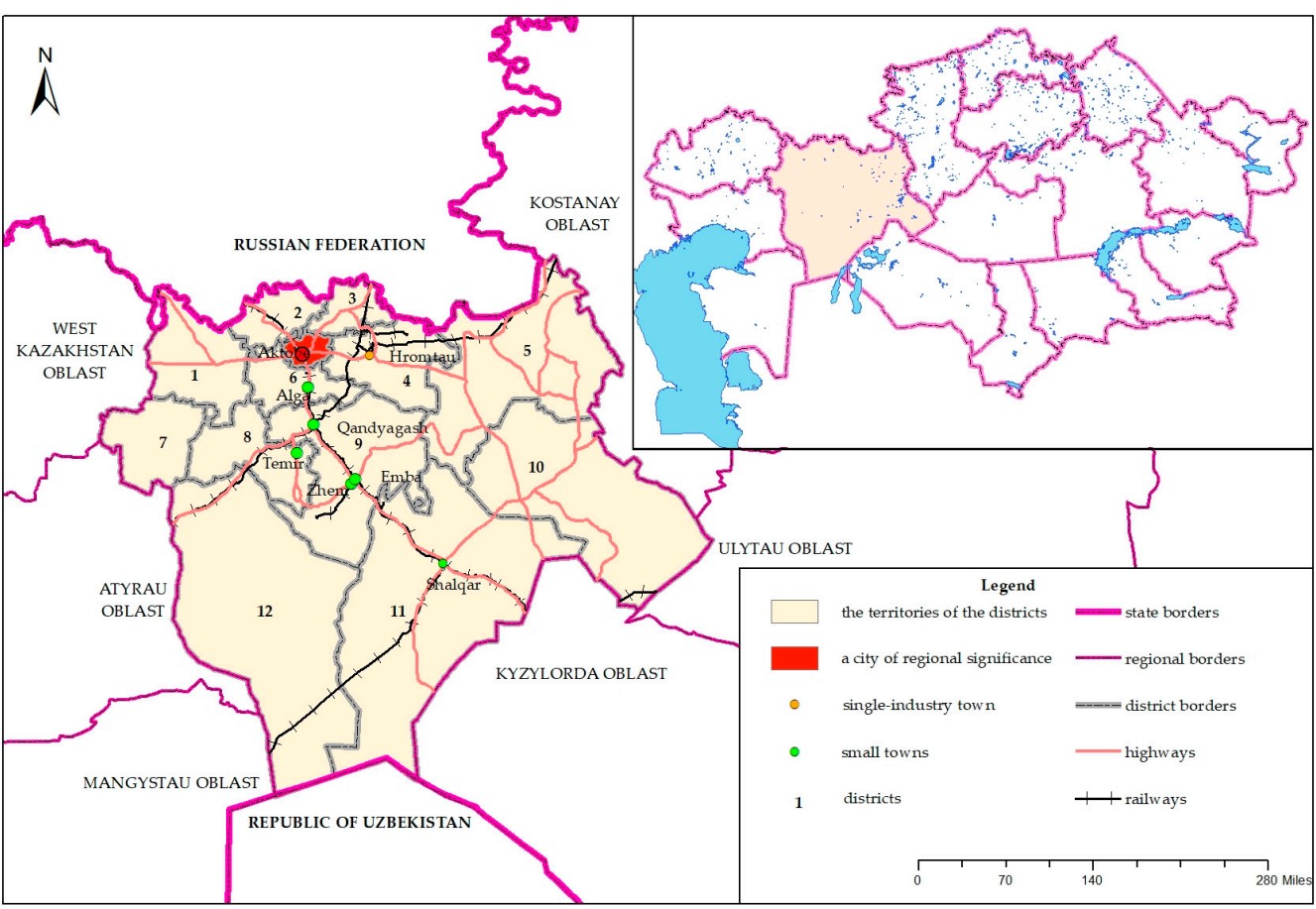

**Figure 1.** Study area. Indicated in numbers on the map: 1. Kobda district; 2. Martuk district; 3. Kargaly district; 4. Khromtau district; 5. Aitekebi district; 6. Alga district; 7. Oiyl district; 8. Temir district; 9. Mugalzhar district; 10. Yrgyz district; 11. Shalkar district; 12. Baiganin district.

In characterizing the climate of the Aktobe oblast, it should be noted that it has a pronounced continental character. The winters here are cold, and the summers are hot and dry. Dry winds and dust storms often occur during the summer, and snowstorms are common in winter. The average temperature in July is around +22.5 °C in the north-west and up to +25 °C in the south-east, while in January it drops to −16 °C and −25.5 °C, respectively. Precipitation varies between about 300 mm in the northwest and from 125 to

200 mm in the center and south of the region. The growing season here ranges from 175 to 190 days a year. The rivers of the Aktobe oblast, belonging to the endorheic basins of the Caspian Sea and small lakes, originate in Mugodzhary.

Aktobe oblast consists of 12 districts. Aktobe oblast is a region with a developed industrial and agricultural base, where agriculture is actively progressing. In parallel with this, measures are being taken to stimulate the development of the tourism sector, including the sector of rural tourism. One of the prospects is the development of rural tourism. Despite all its advantages, rural tourism in the Aktobe oblast is at an early stage of development and has not fully revealed its tourism potential [45].

## 3. Materials and Methods

Assessing the sustainable development of the components of the tourism and recreational potential of rural areas requires an integrated approach and the use of various methods.

In the context of assessing the sustainable development of the tourism and recreational potential of rural areas, the cartographic method can be an effective tool for visualizing various aspects, such as the distribution of natural and cultural resources, infrastructure, environmental conditions and other factors that affect the potential for tourism development.

The use of the cartographic method in the study of the tourism and recreational development of rural areas consists of the following steps.

To compile maps and charts, the necessary data were collected about rural areas and their tourism potential (natural attractions, cultural heritage) [46]. In the course of the study, the indicators of the passenger transport and highways administration of Aktobe oblast (length of highways), the agriculture and land relations administration of Aktobe oblast (area of forests and water bodies), State climate cadastre data [47], the national statistics of the Agency for Strategic Planning and Reforms of the Republic of Kazakhstan were used, as were the following main dynamic indicators of the bureau for Aktobe oblast: the number of people employed in farms [48], the main indicator of the labor market [49], the average monthly salary in Aktobe oblast [50], the number of accommodation places [51], the occupancy of accommodation places at one time (beds) [52], the number of public catering enterprises [53], the indices of industrial production [54], the activity indicators of cultural and recreational facilities [55], and the activity indicators of museums [56].

The use of ArcGis 10.5 tools was the basis for creating maps showing the tourism potential of rural settlements. The development of rural tourism is associated with such a factor as the density of settlements. To determine the density of settlements, the tool in ArcGis 10.5 ArcToolbox "Density" was used. Density estimation is a statistical technique used in ArcGis, often used to determine the location of clustered distribution areas of geographic points. It focuses on the location of a specific point and distributes the properties of the point within a given threshold range. The highest density is in the center and then decreases with distance, gradually approaching zero. The development of rural areas creates a rural color (features of rural life, etc.) in the territories where they are located. The density of settlements is an important factor in the sustainable development of rural tourism. The frequency of concentration of settlements makes it possible to organize rural tourism, and in places where settlements are rare, there are some difficulties in organizing this type of activity (lack of accommodation and catering facilities, low level of infrastructure, etc.) [57].

In addition, during the research, a buffer analysis was performed using the geoprocessing tool of the ArcGis 10.5 program. Buffer analysis is widely recognized as one of the most important spatial analysis tools used to solve proximity problems [58]. Here, buffer analysis was used to analyze the influence of roads, railways, local roads, and rivers on the spatial distribution of rural tourism in settlements of the Aktobe oblast.

It was important to analyze the works of several scientists in identifying areas with high potential for tourism development in rural areas. Among them, the location frequency map of settlements made for Zhejiang and Shaanxi provinces of China, the analysis model of roads through the buffer zone, etc., were reviewed [59].

Several maps were created to visualize the results of the analysis and communicate with stakeholders. The maps help to understand the current situation and potential for tourism development in rural areas. Based on the analysis of the map, it is possible to make decisions about rural tourism development strategies and the measures necessary to achieve sustainable development.

In the context of assessing the sustainable development of the tourism and recreational potential of rural areas, the ranking method was used to determine development priorities and identify the most significant components or factors. The ranking method is an important tool for decision making in situations of limited resources and complex multicriteria problems, making it a useful tool in assessing sustainable tourism development in rural areas.

The process of assessing the sustainable development of rural tourism potential consists of blocks that include the following methods (Figure 2).

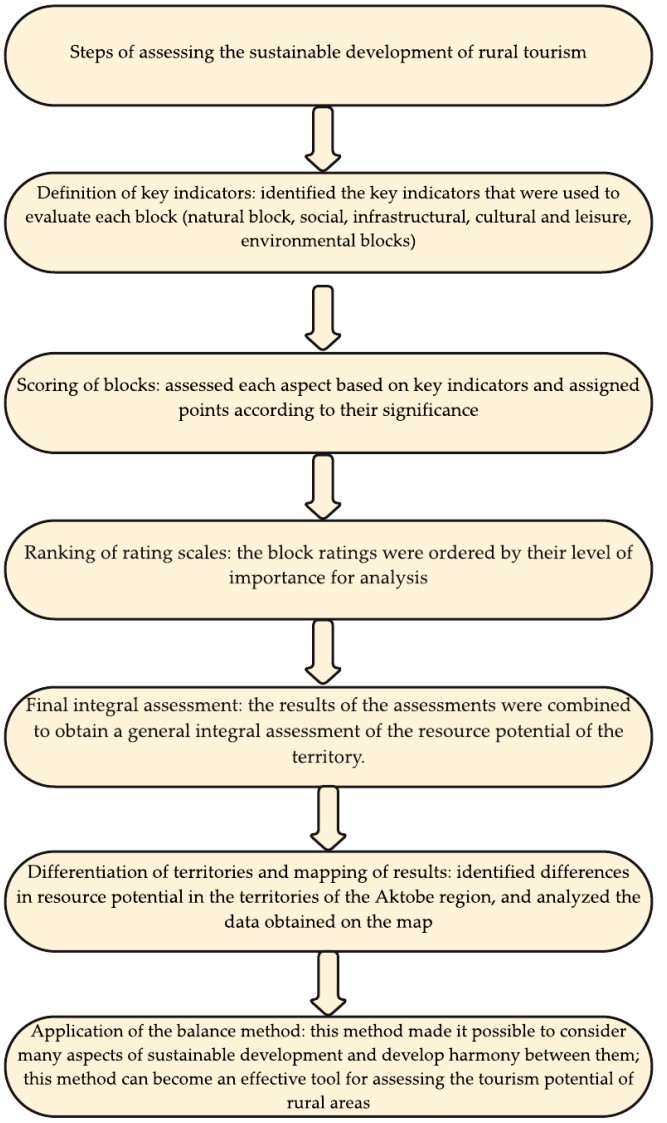

**Figure 2.** Steps of assessing the sustainable development of rural tourism.

In total, 12 districts of the Aktobe oblast were used as territorial units to conduct the study. The analysis methodology is based on four main blocks characteristic of the tourism sector: landscape, social, infrastructure development, cultural and leisure resources, ecological component. A set of factor assessment indicators was determined for each of these blocks, the total number of which is 25 units (Table 1).

Table 1. Criteria for assessing the sustainable development of rural tourism in the Aktobe oblast.

| № | Evaluation Criterion | Point Scale | | | | |
|---|---|---|---|---|---|---|
| | | 1 Point | 2 Points | 3 Points | 4 Points | 5 Points |
| | Natural resources | | | | | |
| | Landscape assessment component k = 1 | | | | | |
| | Share of forest fund lands, % | 0.0001–0.3 | 0.4–0.6 | 0.7–0.9 | 1–1.2 | ≥1.2 |
| | Share of water fund lands, % | 0.0001–0.3 | 0.4–0.6 | 0.7–0.9 | 1–1.2 | ≥1.2 |
| | Number of pyrodic objects, units. | 1–3 | 4–6 | 7–9 | 10–12 | ≥12 |
| | Climate component k = 0.5 | | | | | |
| | Duration of summer comfortable period with t ≥ 15, days. | 30–39 | 40–49 | 50–59 | 60–69 | 70–79 |
| | Duration of the swimming season with average water temperature ≥ 17, days. | 20–29 | 30–39 | 40–49 | 50–59 | ≥60 |
| | Social resources | | | | | |
| | Agriculture k = 1 | | | | | |
| | Number of people employed in peasant or farm enterprises in the region, units. | 100–300 | 301–600 | 601–900 | 901–1200 | ≥1200 |
| | Number of economic entities in agriculture, units. | 50–150 | 151–250 | 251–350 | 351–450 | ≥450 |
| | Population and labor resources k = 0.5 | | | | | |
| | share of employed people in the economically active population, % | 0–20 | 21–40 | 41–60 | 61–80 | 81–100 |
| | index of nominal wages in regions, % | 10–30 | 31–60 | 61–90 | 91–120 | ≥120 |
| | Infrastructure development | | | | | |
| | Service enterprise component k = 1 | | | | | |
| | Number of hotels and similar accommodation facilities, units. | 1–3 | 4–6 | 7–9 | 9–12 | ≥12 |
| | Number of rooms in hotels and similar accommodation facilities, units. | 1–99 | 100–399 | 400–799 | 800–1199 | ≥1200 |
| | Number of public catering places, units. | 1–9 | 10–24 | 25–49 | 50–99 | ≥100 |
| | Industrial component k = 1 | | | | | |
| | Road density per 1000 km$^2$ | 1–19 | 20–39 | 40–79 | 80–119 | ≥120 |
| | indixes of electricity production, gasification, hot water and air conditioning, % | 10–30 | 31–60 | 61–90 | 91–120 | ≥120 |
| | index of services for water supply, collection, treatment and disposal of waste, pollution removal, % | 10–30 | 31–60 | 61–90 | 91–120 | ≥120 |
| | Cultural and leisure resources k = 1 | | | | | |
| | Number of cultural and leisure organizations, units. | 1–9 | 10–19 | 20–29 | 30–39 | ≥40 |
| | Number of cultural events held, units. | 100–299 | 300–499 | 500–699 | 700–899 | ≥900 |
| | Number of museums, units. | 1–2 | 3–4 | 5–6 | 7–8 | ≥9 |
| | Number of excursions held in museums, units. | 100–299 | 300–499 | 500–699 | 700–899 | ≥900 |
| | Number of exhibitions held, units. | 10–30 | 31–60 | 61–90 | 91–120 | ≥120 |
| | Number of historical monuments, units. | 1–3 | 4–6 | 7–9 | 10–12 | ≥12 |
| | Ecological component k = 1 | | | | | |
| | Emissions of pollutants into the atmosphere from permanent sources, tons. | 34–558 | 559–1070 | 1071–6511 | 6512–20,126 | 20,127–53,154 |
| | Level of hydrogen sulfide in the air, MPC (Maximum concentration value mg/m$^3$) | 0.0008–1.0 | 1.1–3 | 3.1–4.9 | 5–8.9 | ≥9 |
| | Quantity of solid mineral deposits, units. | 1 | 2 | 3 | 4 | ≥5 |
| | Unified classification of water quality, class. | 1 | 2 | 3 | 4 | 5 |

To develop a methodology for assessing the sustainable development of rural tourism, methods of complex, statistical, and mathematical analysis were used [60–62]. The use of the balance method in assessing the sustainable development of rural tourism made it possible to quantitatively express the specifics of the tourism sector, to consider in detail the diversity of the resource base, the structure of available opportunities for the development of rural tourism and recreation, to develop a basis for a comparative analysis of the tourist opportunities of the territory, and to determine the priority areas of tourist and recreational activities in each oblast.

To determine the sustainable development of rural tourism, the following formula have been developed:

$$I_{sd} = (I_{nr} + I_{sr} + I_{id} + I_{cr}) - I_{eco}$$

$I_{sd}$—rural tourism sustainable development index;
$I_{nr}$—natural resource index;
$I_{sr}$—social resource index;
$I_{id}$—infrastructural development index;
$I_{cr}$—cultural resources index;
$I_{eco}$—ecology index.

To calculate indicator values, the initial indicators are normalized. The standardization method is based on determining the "most favorable" and "least favorable" values of each indicator for a set of territories. The formula for normalizing the values of indicators that have a positive impact on the quality of development of a certain assessment block has the form:

$$I_{norm} = \frac{I_{max^i} - I_{min^i}}{n^i}$$

$I_{norm}$—standardized indicators included in the group of natural, infrastructural, social and cultural indices;
$I_{max^i}$—maximum value of the i indicator;
$I_{min^i}$—minimum value of the i indicator;
$n^i$—number of I indicators in the component.

To determine the sustainable, balanced development of the oblast, we will evaluate each indicator separately.

$$I_n = I_{norm} \times \sum k_n \times a_n$$

$I_n$—development indices of natural, infrastructural, social and cultural spheres;
$k_n$—the significance coefficient of a specific component in the block evaluation structure;
$a_n$—value of block component evaluation parameters.

## 4. Results and Discussion

After applying the mapping method to assess the sustainable development of the tourism potential of rural areas, several results were obtained that will be discussed with stakeholders and used for decision-making. The results and discussion included the following aspects.

In the development of rural tourism, the variety of services that meet the needs of visitors is directly related to the frequency of settlement locations. In this regard, low indicators were determined in the process of differentiating the distribution frequency of settlements in the oblast. The spatial distribution density of the village in the Aktobe oblast is 80.7 per 300.629 km$^2$. The highest density of settlements varies from 53.8 to 80.7 and is typical for areas located in the northern part of the oblast such as the following: the Martuk, Khromtau, Kargaly, and Kobda districts. The natural environment, transportation, and economic development have shaped the areas of high population density. The Baiganin, Irgiz, Shalkar, and Aitekebi districts are characterized by a low density of settlements. The low-density index varies from 17.7 to 35.7 (Figure 3).

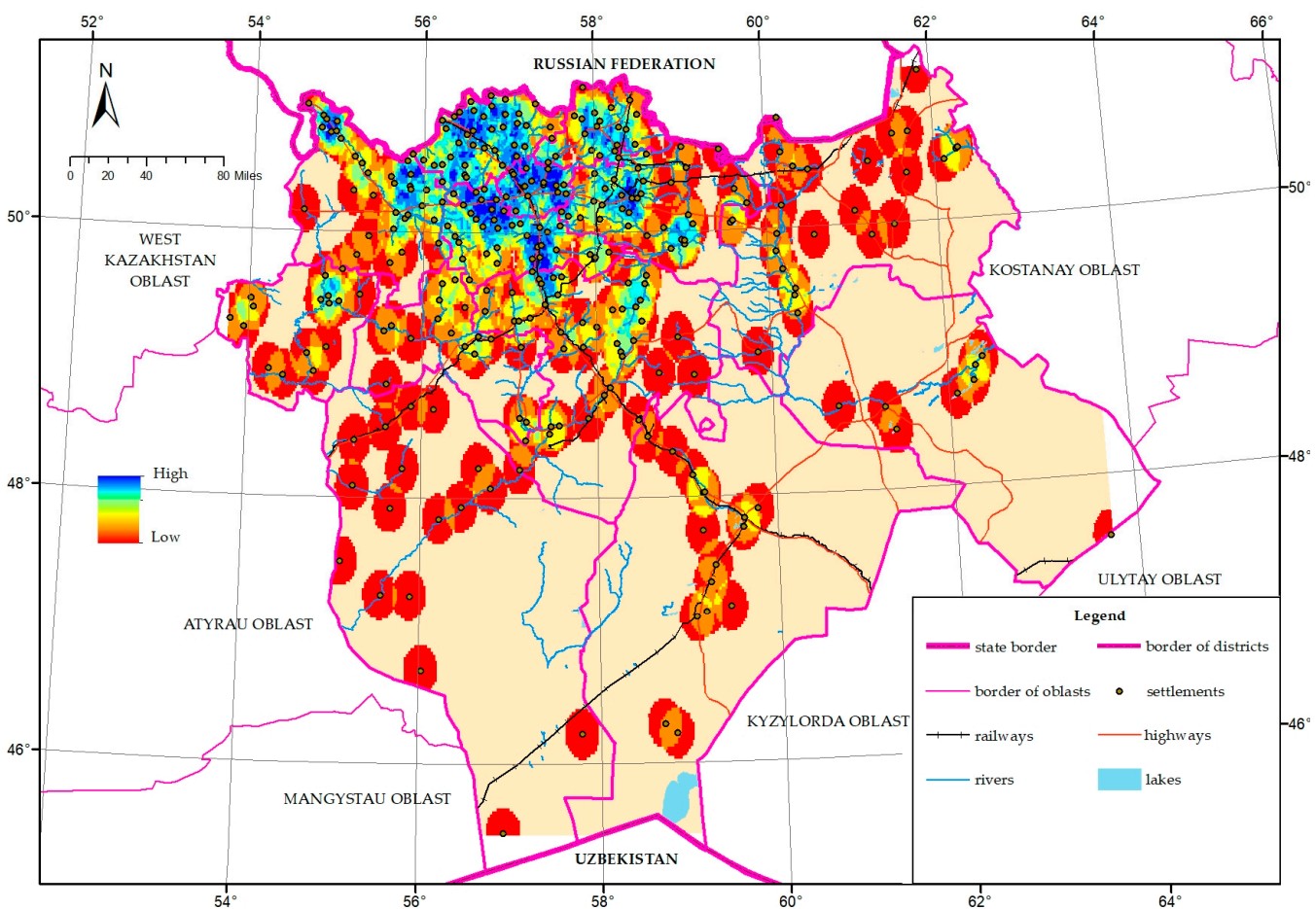

**Figure 3.** Distribution of village density.

The cores of the ecological framework identified on the territory of the Aktobe oblast form rows stretching from north to south and from west to east; they cover all landscape zones, providing the possibility of conservation and the safe migration of species in the latitudinal and meridional directions.

Figure 4 also suggests targeted protection areas that are valuable for preserving individual components of nature: unique landforms, azonal and intrazonal landscapes. Almost all large sandy massifs and natural forests in the oblast fall into this category since they are relics of previous geological eras and include various biotopes that combine zonal and azonal elements.

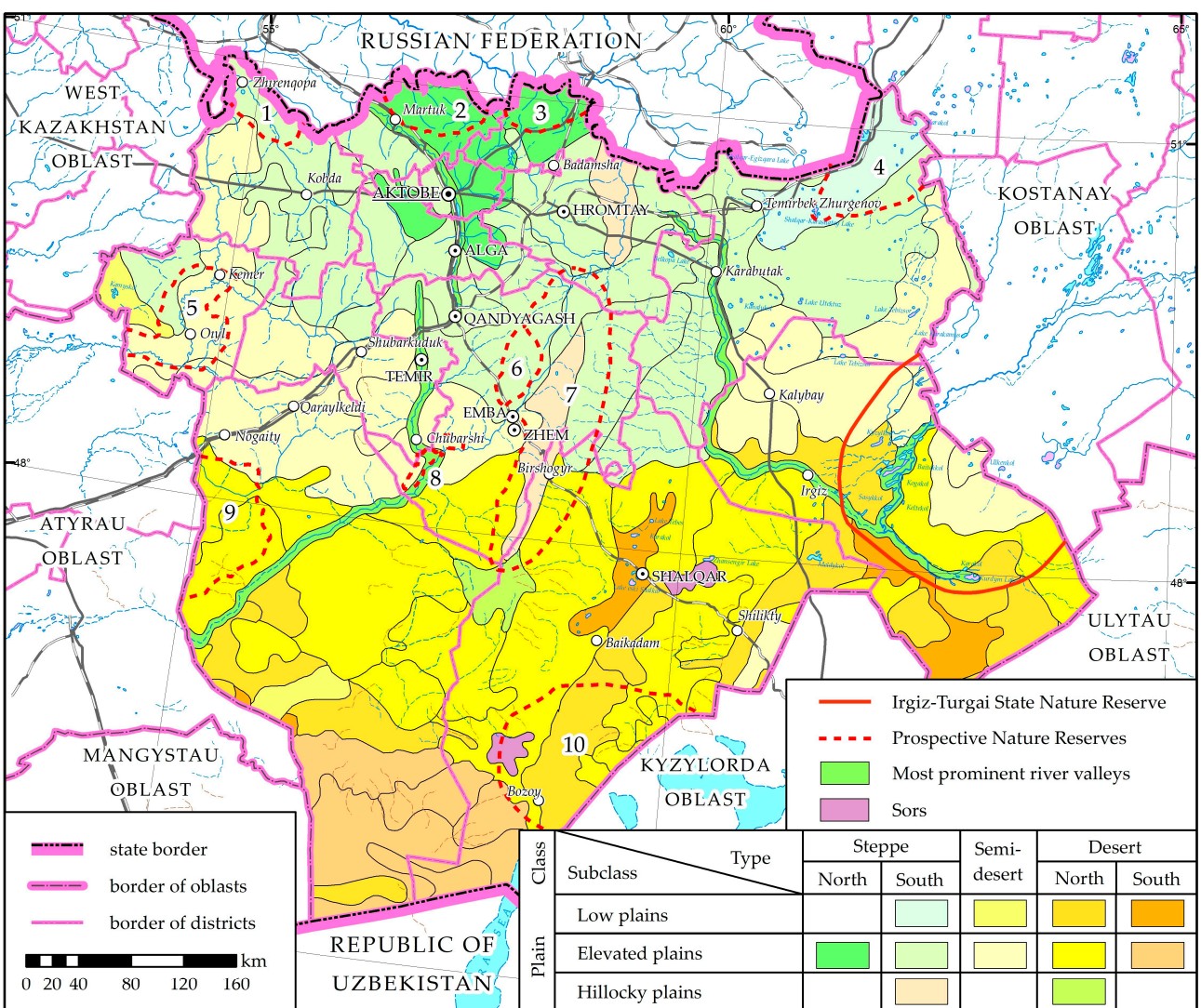

**Figure 4.** Ecological framework of the Aktobe oblast. Indicated in numbers on the map: 1. Kobda; 2. Martuk; 3. Ebita; 4. Ozernyi; 5. Oiyl; 6. Orkash; 7. Mugalzhar; 8. Kokzhide–Kumzhargan; 9. Sagyz; 10. Grand Borsyk.

As a result of a comprehensive assessment, territories with well-preserved steppe ecosystems were selected, suitable for organizing specially protected natural areas (SPNA), the creation of which will be aimed at preserving all elements of nature and biological diversity because of preserving the habitat of living beings.

The established natural harmony of landscapes performing various ecological functions has formed an ecological framework, and in the case of creating a protected area here, we will achieve high results in preserving biodiversity for the northern part of Aktobe oblast. It can be formed in the west of the oblast based on the harmony of the regional landscapes of the dry and desert steppe intersected with the intrazonal plain landscapes of the Oyil River and the azonal landscapes of the Barkyn sand basin [63–65].

It is located within the zonal landscapes of the steppe-turned-desert as the basis of the future ecological network. This is the Yrgyz–Torgai state natural reserve, which consists of two territories with the Torgai state nature reserve located between them. Yrgyz, Torgai, etc., have a lot of meadows and old lakes, marshes and bay meadows. The intrazonal landscape of regional river valleys form the food base of migration routes (ecological corridors) for birds from India, Pakistan, and North African wintering areas.

In order to preserve the unique biotopes found in the Kishi and Ulken Borsyk sand basins, the Akkumsagyz sand, the chalky plateau of Donyztau ridges, and the Aktolagay

ridges, it is necessary to organize protection properly. In the active processes of subsidence and desertification in the vicinity of the Aral region, these territories are polygons for forming the modern terrain and their anthropogenic transformation.

The environmental condition of Aktobe oblast was considered in terms of air pollution, the amount of hydrogen sulfide (hydrogen sulfide) in the air, the presence of solid mineral deposits in the regions, and the quality of water resources (Figure 5).

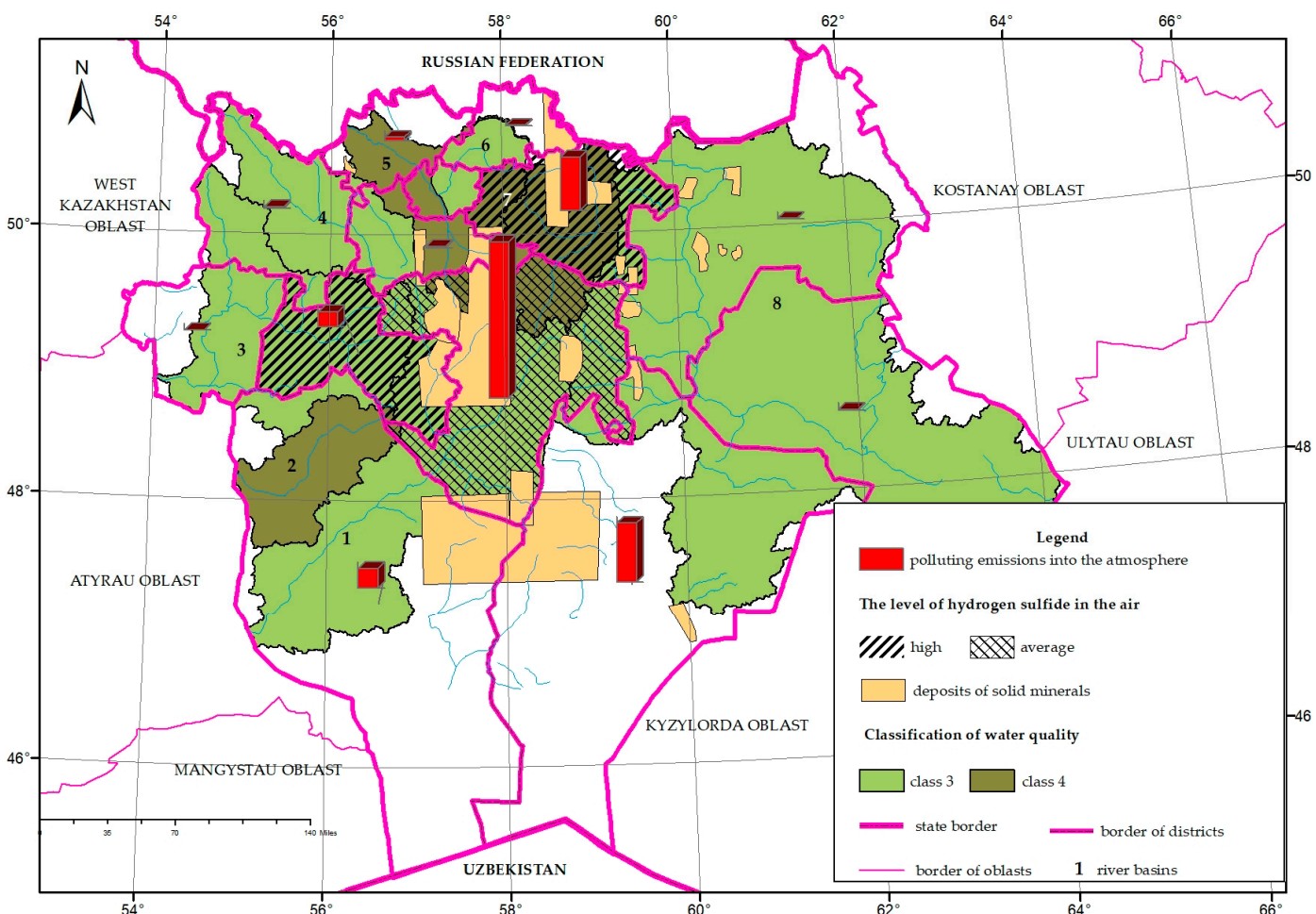

**Figure 5.** The state of the environment of the Aktobe oblast for the development of rural tourism. Indicated in numbers on the map: 1. Basin of the Zhem River; 2. basin of the Sagyz River; 3. basin of the Oyil River; 4. basin of the Kobda River; 5. basin of the Elek River; 6. basin of the Kargaly River; 7. basin of the Or River; 8. basin of the Yrgyz River.

Regarding atmospheric pollution, the Mugalzhar (53,157 tons) and Khromtau (18,056 tons) districts showed indicators. The amount of hydrogen sulfide in the atmosphere was recorded in three districts of the region. The highest indicator is 4.5 mg/m$^3$ in the Khromtau district and 2.3 mg/m$^3$ in the Temir district. In the Mughalzhar district, the amount of hydrogen sulfide is the lowest—1 mg/m$^3$ [66,67]. The level of air pollution in these areas depends on the abundance of mineral deposits. Mining deposits of solid minerals are also found in the Alga, Martuk, Kargaly, Aitekebi, Baiganin, Shalkar districts [68]. The quality of the water resources of Aktobe oblast was considered based on eight main water basins [69]. The main regulatory document for assessing the water quality of the water bodies in the Republic of Kazakhstan is the "Unified system of classification of water quality in water bodies". The unified system for classifying water quality in water bodies is divided into six water use classes with a gradual transition from class 1 of "best quality" water to class 6 of "worst quality". Each water use class is characterized by its water use

category depending on the formed ecological potential of the water body. For recreation, water classes 1, 2, 3 are recommended [70]. The surface water in Aktobe oblast belongs to the fourth class. Only the Kargaly and Emba rivers are recorded as belonging to class ≥ 3. According to this class indicator, only river basins of category ≥ 3 are suitable for recreational use.

According to the information bulletin on the state of the environment of the Aktobe oblast, the National Hydrometeorological Service of Kazakhstan, 2023, a map analysis can help identify areas with high potential for tourism development that may have gone unnoticed until now. This includes areas with unique natural resources, cultural heritage, social resources and infrastructural development that can be developed, taking into account the principles of sustainable development (Figure 6).

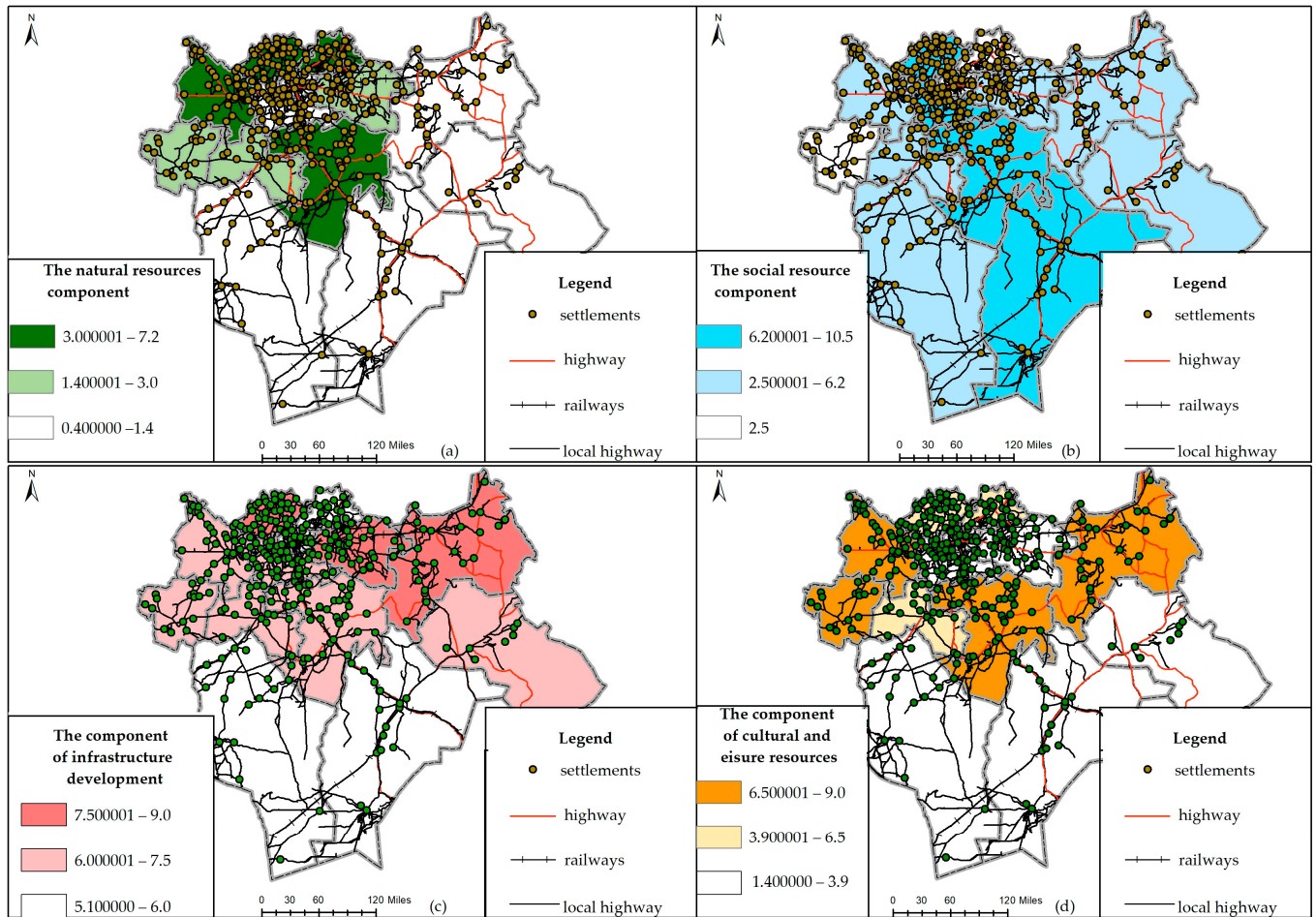

**Figure 6.** Positive components of assessing the sustainable development of rural tourism: (**a**) natural resources; (**b**) social resources; (**c**) infrastructural development; (**d**) cultural and leisure component.

The highest index for the assessment of the natural resources component is $-I_n = 7.2$ in the districts of Kargaly and Martuk. If the Kargaly district differs from other districts by the number of natural landscape places (Sarkyrama, Ebita, Shaushke, etc.), the Martuk district was highly evaluated in the mentioned parameter by the share of forests and water bodies. The parameter of the social development of the districts was evaluated according to the number of owners under the age of 35 in the field of agriculture, the number of people employed in the peasant or farmer farms of the district, the number of economic entities in agriculture, the share of the number of employed people from the economically active population, and the indicators of the nominal wage index in the districts. According to this parameter, the scores of Alga, Martuk ($I_n = 10.5$) and Mugalzhar ($I_n = 9.6$) belong to these districts. The largest number of farms and the highest level of employment in these

farms were recorded in these districts. The Khromtau, Aitekebi, Alga, Kobda, Mugalzhar districts showed a high infrastructure development. These districts were rated relatively high in terms of location, number base, and the number of public catering establishments. The Mughalzhar and Kobda districts had a high index in terms of cultural and recreational resources. The number of cultural and recreational places and events held in these districts is relatively high.

The assessment of the sustainable development of rural tourism of the Aktobe oblast was based on five factors: natural resources, the social features of rural settlements, infrastructure development and cultural and recreational resources, and ecological factors. As a result of the assessment, the following indicators were determined. The Martuk, Kobda, Kargaly districts have been identified as areas with a high potential for rural tourism development according to the principle of sustainable development. The Baiganin, Alga, Oyil, Mughalzhar, Shalkar, Yrgyz districts showed an average indicator, indicating a need for further development to fully tap into their rural tourism potential. The Aitekebi, Khromtau, and Temir districts had a low indicator, highlighting the importance of concerted efforts to boost their rural tourism initiatives (Table 2).

**Table 2.** Assessment of the potential for sustainable development of rural tourism in Aktobe oblast.

| | Natural Resources | Social Resource | Infrastructure Development | Cultural and Leisure Resources | Environmental Factor | Rural Tourism Sustainable Development Index |
|---|---|---|---|---|---|---|
| Alga | 1.1 | 10.125 | 7.5 | 1.4 | 6 | 14.125 |
| Aitekebi | 1.3 | 5.75 | 8 | 1.76 | 10 | 6.81 |
| Baiganin | 1.4 | 5.75 | 6 | 8.5 | 3.5 | 18.15 |
| Kargaly | 7.2 | 2.375 | 5.5 | 6.5 | 2 | 19.575 |
| Kobda | 4.8 | 6.25 | 7.5 | 9 | 3.75 | 23.8 |
| Martuk | 7.2 | 10.25 | 8 | 5.5 | 5.25 | 25.7 |
| Mugalzhar | 6.8 | 8.75 | 7 | 9 | 15 | 16.55 |
| Oiyl | 3 | 2.375 | 7 | 8.4 | 3.75 | 17.025 |
| Temir | 2.5 | 5.75 | 6.5 | 1.28 | 6.5 | 9.53 |
| Khromtau | 1.95 | 5.75 | 9 | 1.28 | 8 | 9.98 |
| Shalkar | 1.2 | 8.75 | 5.1 | 3.9 | 2.75 | 16.2 |
| Yrgyz | 0.4 | 2.375 | 6.5 | 4.2 | 3 | 10.475 |

In terms of the infrastructure condition assessment, cartographic analysis made it possible to assess the availability of tourism infrastructure, such as hotels, campsites, cafes and other facilities necessary to meet the needs of tourists.

There are 102 locations in Aktobe oblast and they can accommodate 6005 visitors at a time. If we classify the number of these locations by districts, the largest number of locations belongs to the Khromtau district (10 locations served about 3900 visitors in 2023). There is a shortage of accommodation in the Alga, Oyil, Baiganin districts. The highest number of catering establishments is in the Mughalzhar district (there are 86 public catering establishments (Figure 7)).

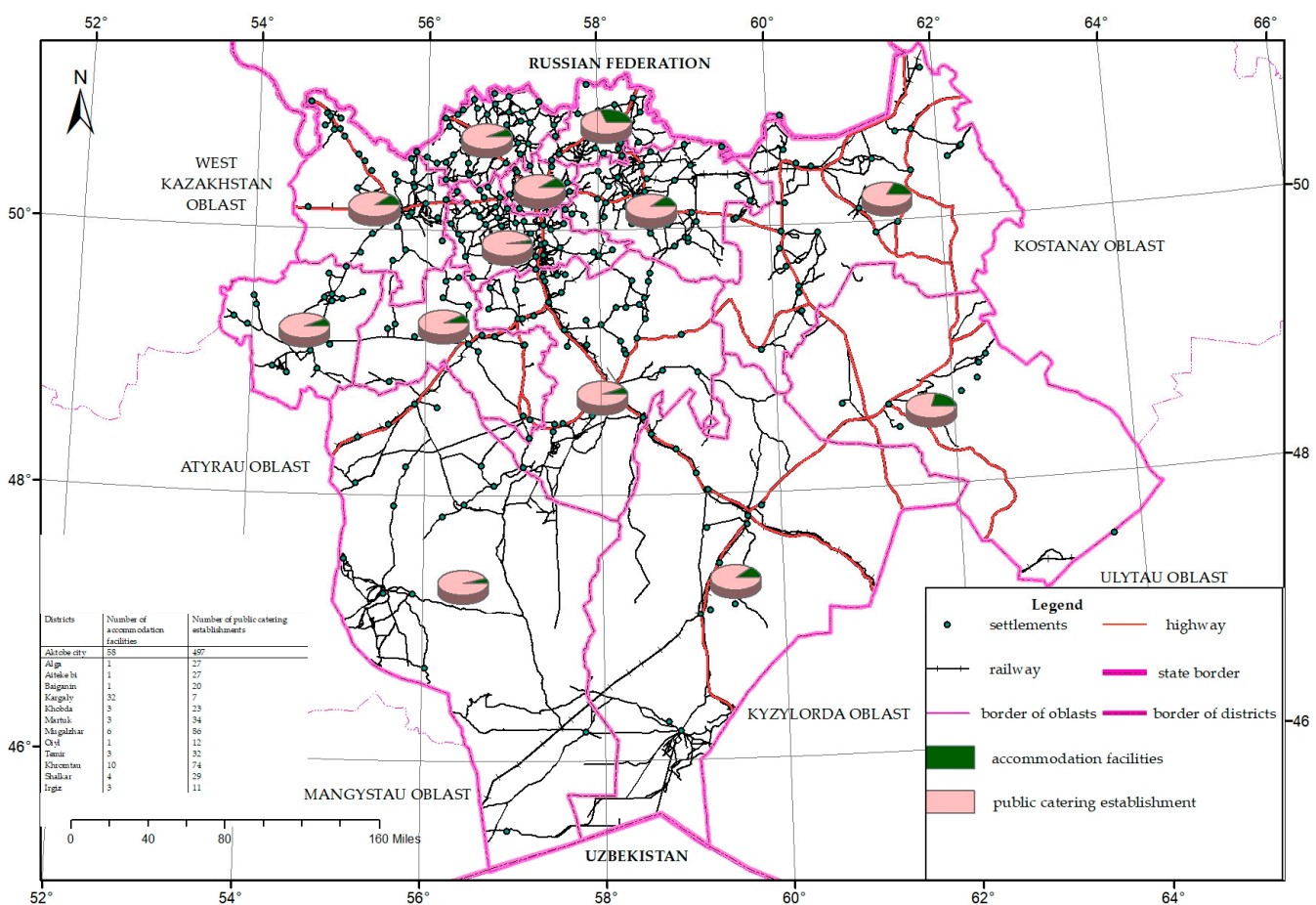

**Figure 7.** Number of accommodation facilities and public catering public catering establishments in the oblast.

The level of transport infrastructure is the leading main indicator of meeting the demands of visitors. In addition, the location of settlements in river valleys contributes to increased activities offered to visitors (fishing services, bathing, etc.). In this regard, potential settlements located on highways, railways and river valleys were identified during the research. Buffer analysis was used to study the influence of transport modes: highways, highways and railways, and rivers on the spatial layout of rural tourism in rural areas. The buffer radius of main roads, secondary roads, railways and rivers has been set at 10 km. Once the buffer range of roads was obtained, a map of the distribution of buffer zones within various roads was generated. There are 317 settlements in the Aktobe oblast. Of these, 140 (44.1%) settlements are located in the highway buffer zone, 88 (27.7%) settlements are within the railway buffer zone, 273 (86.11%) within the river branches and almost all the villages were located in the buffer zone of secondary roads. This suggests that road transport infrastructure and river massifs spatially shape the layout of settlements for rural tourism development (Figure 8).

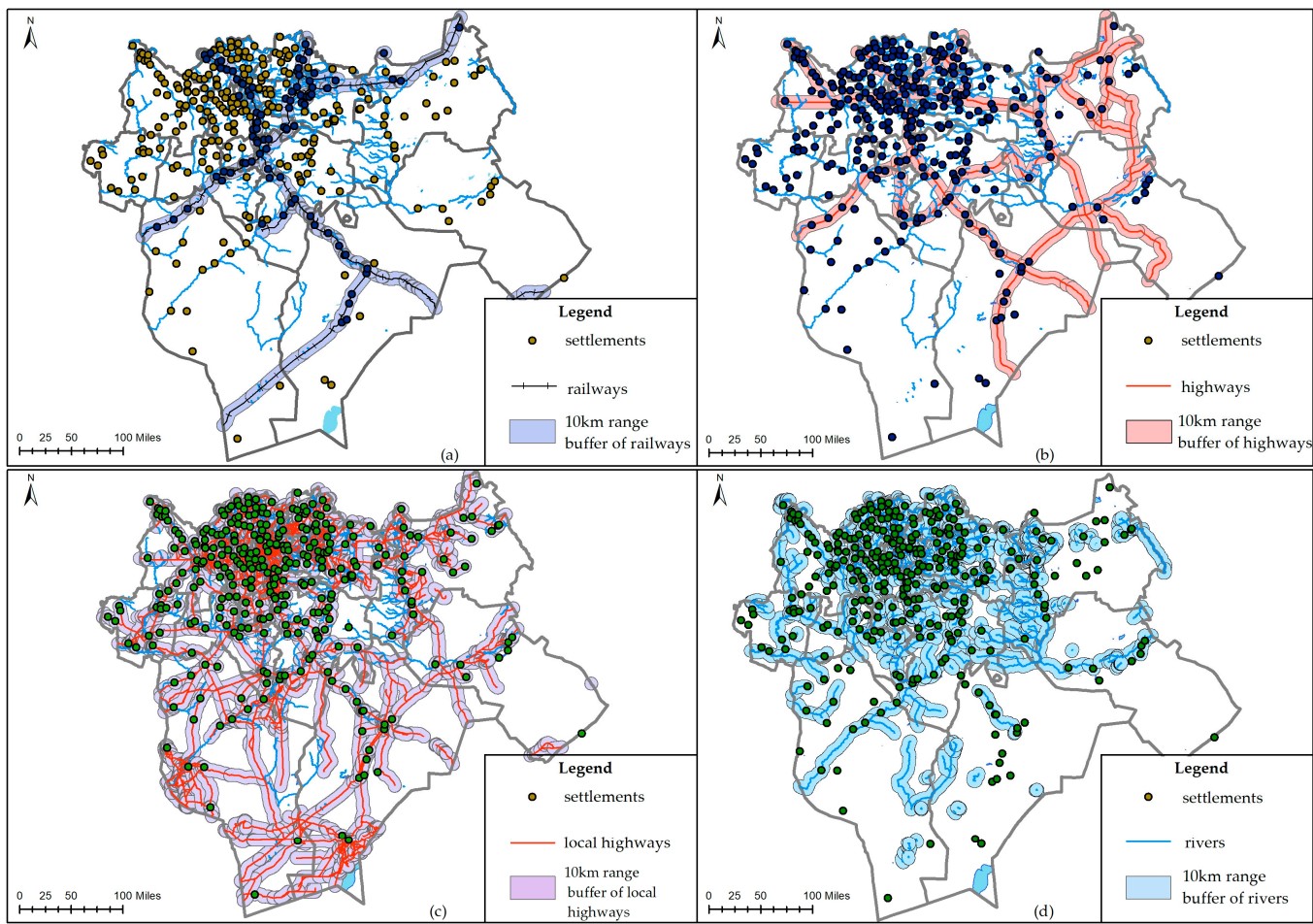

**Figure 8.** Buffer zones of rural areas of the Aktobe oblast: (**a**) railways; (**b**) highways; (**c**) local roads; (**d**) rivers.

The development of rural tourism is closely related to gastronomic characteristics and the types of agricultural services of the rural population. In the regions of the Aktobe oblast, peasant farms are engaged in producing meat and milk as the main activity. However, they are not included in the process of providing tourism and excursion services to the population everywhere, and only 20% are engaged in searching for an additional source of income through attracting tourists. In this regard, the development of rural tourism in the regions can be associated with Zhailau tourism. Zhailau is a place convenient for summer cattle keeping. Zhailau tourism includes a special synthesis of rural and ethnotourism, which is typical for all regions of Kazakhstan, including the Aktobe region. The characteristic lowland landscapes are considered unique compared to grasslands in other countries such as Kyrgyzstan. An attractive feature of Zhailau tourism is the healing grace in the summer and the traditional flavor of nomadic life: fresh air, spring water, and organic food from meat and dairy products. The natural attractions of Zhailau tourism are harmoniously combined with the historical and cultural heritage, creating a unique aura of immersion in the original culture of nomads: an overnight stay in a yurt, folk songs, national games and rituals, horseback riding, master classes on folk crafts and national handicrafts made of wool and leather and other local materials. The tourists of Zhailau tourism are proposed to be classified according to their affiliation with the main purpose of the tour as follows:

1.  Tourists whose main purpose is koumiss treatment (koumiss treatment is the general name of the healing process that uses fresh mare's milk (saumal) and koumiss) and

shubat treatment (shubat (camel milk) can be recommended to people suffering from many diseases);

2.  Transit tourists stopping overnight on their way to other regions of the country or neighboring countries;

3.  Tourists relaxing on the river banks and briefly visiting Zhailau for an excursion.

For the sustainable development of jailau tourism, it is necessary to constantly cultivate summer pastures by all interested parties in the country: to maintain them in an attractive condition for tourists, both in natural and social terms.

In the process of indexing based on the balance method, problem areas were identified among the districts of the oblast; among them are the Khromtau district, Aitekebi district, and Temir district. Since the Khromtau district is an industrial zone, environmental pollution indices showed a high level. A small number of landscaped areas and a low level of infrastructure were found in Aitekebi and Temir districts. These indicators have an inhibitory effect on the development of rural tourism in these areas. Problem districts were indexed by parameters of natural resources, social situation, infrastructural development, cultural–recreational resources, and environmental resources, showing low results (Figure 9).

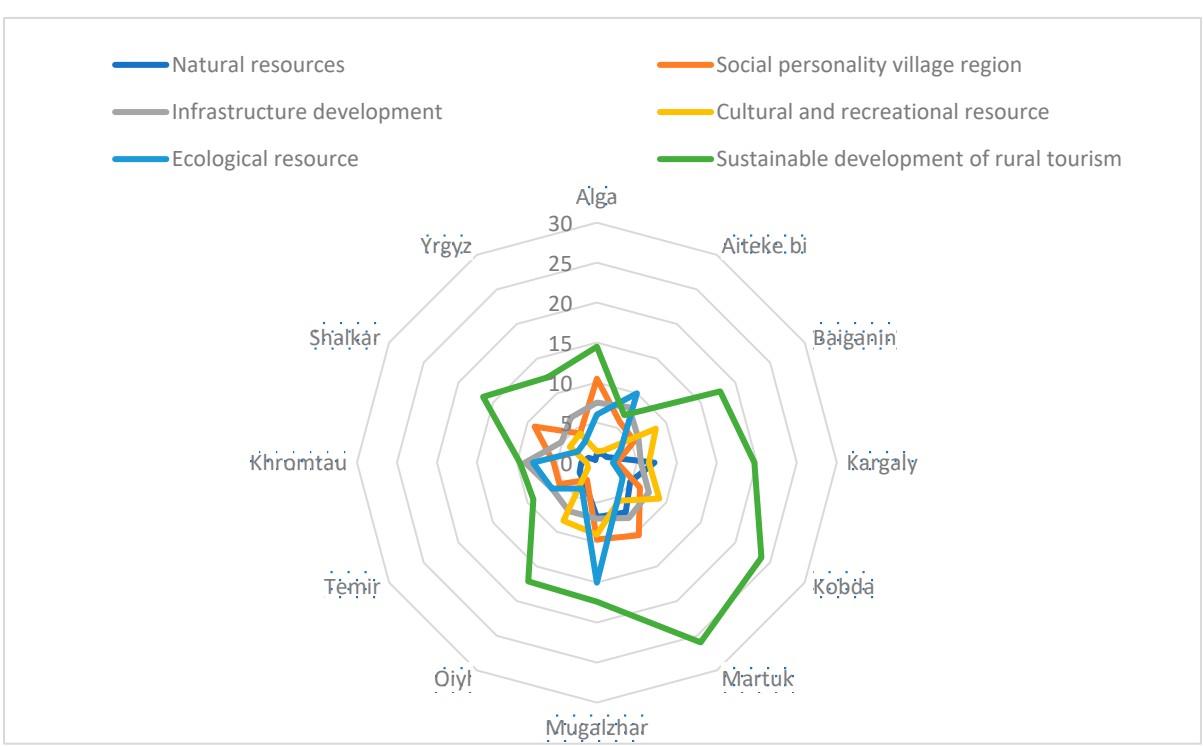

**Figure 9.** Identification of problem areas for the development of rural tourism in Aktobe oblast.

The study of rural tourism as a socio-economic phenomenon allows us to make the following conclusions:

Rural tourism develops selectively and slowly in the socio-geographical space and is initiated mainly through private initiative. Positive projects are observed in the northern oblasts of Martuk, Kargaly and Kobda.

In the Aktobe oblast, rural tourism development models are formed on the basis of the resource specifics of the territory. The development of rural tourism must take into account the unique characteristics of each oblast.

Rural tourism is complex, fitting into territorial and economic–geographical constructs.

The development of rural tourism in the Aktobe oblast is based on various models, including development based on peasant farms, private farms, horse breeding centers and others.

The multi-resource nature of the territory of the Aktobe oblast allows the development of rural tourism in various directions, considering the specifics of agricultural sectors.

Regional policy for the development of rural tourism should consider the characteristics of the territory and the specifics of various types of rural areas.

The understanding and development of rural tourism requires a spatial analysis of the resource base at the regional level, which will become the basis for developing concepts for the development of rural tourism in the Aktobe oblast.

The technology for the development of rural tourism is associated with the general economic features of the territory and the attractiveness of the rural landscape, which requires the correct zoning and consideration of the specifics of each area.

## 5. Conclusions

Assessing the accessibility of tourism infrastructure and conducting an analysis made it possible to identify vulnerable areas and risks associated with the development of rural tourism, and to identify areas with high potential for the development of tourism in the Aktobe oblast. This will help develop measures to minimize risks and maximize socio-economic benefits while complying with the principles of sustainable development.

The results obtained highlight the importance of rural areas in the Aktobe oblast as tourist destinations and the need to develop comprehensive development strategies that consider environmental, economic and socio-cultural aspects. Continued collaboration with stakeholders and taking action to realize identified opportunities are key steps towards sustainable tourism development in rural areas of the Aktobe oblast.

The use of cartographic and balance methods in assessing the sustainable development of the tourism potential of rural areas allows us to consider an integrated approach to tourism development, considering environmental, economic, and sociocultural aspects. This allows us to not only identify key factors and problems, but also to propose constructive strategies and recommendations for rural tourism development in accordance with the principles of sustainability.

**Author Contributions:** Conceptualization, A.S., M.O. and K.S. (Kuat Saparov); methodology, A.S., M.O. and K.S. (Kuat Saparov); software, M.O. and A.S.; validation, K.S. (Kairat Saginov) and A.Y.; formal analysis, A.S. and M.O.; investigation, A.S., M.O., K.S. (Kuat Saparov), K.S. (Kairat Saginov), A.Y. and G.A.; resources, K.S. (Kairat Saginov), A.Y. and G.A.; data curation, A.S. and M.O.; writing—original draft preparation, A.S. and M.O.; writing—review and editing, K.S. (Kuat Saparov); visualization, K.S. (Kairat Saginov), A.Y. and G.A.; supervision, A.S. and M.O.; project administration, A.S. and M.O.; funding acquisition, A.S., M.O., K.S. (Kuat Saparov), K.S. (Kairat Saginov), A.Y. and G.A. All authors have read and agreed to the published version of the manuscript.

**Funding:** This research received no external funding.

**Institutional Review Board Statement:** Not applicable.

**Informed Consent Statement:** Not applicable.

**Data Availability Statement:** The data that support the findings of this study are available upon reasonable request from the authors.

**Conflicts of Interest:** The authors declare no conflicts of interest.

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
