# Peer review of "Assessment for the Sustainable Development of Components of the Tourism and Recreational Potential of Rural Areas of the Aktobe Oblast of the Republic of Kazakhstan"

_sustainability, doi:10.3390/su16093838_

Round 1

Reviewer 1 Report

Comments and Suggestions for Authors

The article „Assessment for the sustainable development of components of the tourism and recreational potential of rural areas of the Aktobe oblast of the Republic of Kazakhstan“ is very interesting, especially with regard to the sustainable development of rural tourism in the Aktobe oblast of the Republic of Kazakhstan. I think that the article could be improved in some parts.

I suggest that the authors link the article more closely to previous research on the sustainable development of rural tourism. In the theoretical part of the article, more recent sources on sustainable rural tourism must be included. The research questions need to be explained.

In my opinion, the title of the article is too long, so I would suggest the authors to shorten it. In the conclusion research limitations and recommendations for future research should be presented. The conclusion should be better explained.

Author Response

We would like to express our sincere gratitude to you for your important research comments, which helped to enrich and improve the content of our work. Their value lies in the fact that they helped us deepen our understanding of the subject of study, identify additional aspects of the problem, and also suggest additional methods for data analysis. This made our research more comprehensive and informative. We wish you all the best and creative success.

Reviewer’s comment:

  1. The article „Assessment for the sustainable development of components of the tourism and recreational potential of rural areas of the Aktobe oblast of the Republic of Kazakhstan“ is very interesting, especially with regard to the sustainable development of rural tourism in the Aktobe oblast of the Republic of Kazakhstan. I think that the article could be improved in some parts. I suggest that the authors link the article more closely to previous research on the sustainable development of rural tourism. In the theoretical part of the article, more recent sources on sustainable rural tourism must be included. The research questions need to be explained.

Response to reviewer:

Corrected according to the comments:

Thanks to the ecological and tourism potential, which is part of the natural resource potential, the territories fulfill their socio-cultural load, which contributes to their sustainable development. The need to study the problem of formation, development and assessment of the ecological and tourism potential of the region is determined not only by the need to diversify the use of resource opportunities, but to a greater extent by the turbulent state of the social and labor sphere. In addition, the lack of general methodological approaches to assessing the ecological and tourism potential does not contribute to its effective use and, as a consequence, to the stable socio-economic development of territories. The study of the natural prerequisites for the development of the tourism sector is traditionally the first stage in assessing the territory, since natural resources are one of the factors that predetermine its use.

The countryside, with its attractive appearance and the opportunity for city resi-dents to enjoy nature and a healthy lifestyle, satisfies their psychological needs and becomes a valuable resource in modern society. The commodification of this space, in-fluenced by consumer culture, increases its value as landscape, environment and products. However, this process can also lead to the abstraction and symbolization of rural space. Problems associated with excessive commodification are particularly felt in rural areas near large cities, reflecting a shift from overt to hidden forms of loss of rural identity .

Challenges relate to the unavailability of resources such as quality labor and in-vestment, as well as the inability to use local resources to develop rural tourism. The lack of planning and government support demonstrates insufficient attention to the resource approach in research related to rural tourism .

More in-depth research can be undertaken using a resource-based approach to understand the dynamic capabilities of rural destinations and identify how internal and external resources can be effectively identified, mobilized, harnessed and main-tained. This will maximize the benefits of rural tourism.

Modern changes in the preferences of consumers of recreational services, as well as the growth of environmental consciousness of vacationers, create demand for new forms of tourism products that use alternative recreational resources, including the resources of rural areas. In Kazakhstan, where agriculture is a diversified production, there is potential for the development of rural tourism as an additional factor in stimulating tourism in general. The culture, traditions and hospitality of the Kazakh people can give a new impetus to the development of tourism in rural areas, in accordance with the principles of sustainable development.

Aktobe oblast is one of the regions of Kazakhstan where rural tourism can become a priority sector of the economy. However, the territory of the region is characterized by insufficient recreational and geographical knowledge, which is one of the main factors hindering the development of the tourism industry, which is at the initial stage of formation. In this regard, the need arose for a comprehensive assessment of the environmental and tourism potential of the Aktobe region and determination of its regional specifics and prospects for use.

Added more new sources to the article:

  1. Guizzardi, A., Stacchini, A., & Costa, M. (2021). Can sustainability drive tourism development in small rural areas? Evidences from the Adriatic. Journal of Sustainable Tourism, 30, 1280 - 1300. https://doi.org/10.1080/09669582.2021.1931256.
  2. He, Y., Wang, J., Gao, X., Wang, Y., & Choi, B. (2021). Rural Tourism: Does It Matter for Sustainable Farmers’ Income?. Sustainability. https://doi.org/10.3390/su131810440.
  3. Jing, W., Zhang, W., Luo, P., Wu, L., Wang, L., & Yu, K. (2022). Assessment of Synergistic Development Potential between Tourism and Rural Restructuring Using a Coupling Analysis: A Case Study of Southern Shaanxi, China. Land. https://doi.org/10.3390/land11081352.
  4. Safa, R., Yasouri, M., & Hesam, M. (2021). Effects of Tourism on Sustainable Rural Livelihoods (Case Study: Saravan, Rasht County, Iran). Journal of Research and Rural Planning, 10, 1-19. https://doi.org/10.22067/JRRP.V10I3.85201.
  5. Škatarić, G., Spalević, V., Popović, S., Perošević, N., & Novićević, R. (2021). The Vernacular and Rural Houses of Agrarian Areas in the Zeta Region, Montenegro. Agriculture. https://doi.org/10.3390/agriculture11080717.

Reviewer’s comment:

  1. In my opinion, the title of the article is too long, so I would suggest the authors to shorten it.

Response to reviewer:

Dear reviewer, we agree with your comments. In accordance with your comments, we have shortened the title of the article : Assessment for the sustainable development of components of the tourism and recreational potential of rural areas of the Aktobe oblast of the Republic of Kazakhstan

Reviewer’s comment:

  1. In the conclusion research limitations and recommendations for future research should be presented. The conclusion should be better explained.

Response to reviewer:

The conclusion of the article was changed in accordance with the comments:

Assessing the accessibility of tourism infrastructure and conducting analysis made it possible to identify vulnerable areas and risks associated with the development of rural tourism, as well as to identify areas with high potential for the develop-ment of tourism in the Aktobe oblast. This will help develop measures to minimize risks and maximize socio-economic benefits while complying with the principles of sustainable development.

The results obtained highlight the importance of rural areas in the Aktobe oblast as tourist destinations and the need to develop comprehensive development strategies that take into account environmental, economic and socio-cultural aspects. Continued collaboration with stakeholders and taking action to realize identified opportunities are key steps towards sustainable tourism development in rural areas of the Aktobe oblast.

The use of cartographic and balance methods in assessing the sustainable devel-opment of the tourism potential of rural areas allows us to consider an integrated ap-proach to tourism development, taking into account environmental, economic and so-ciocultural aspects. This allows not only to identify key factors and problems, but also to propose constructive strategies and recommendations for the development of tour-ism in accordance with the principles of sustainability.

Reviewer 2 Report

Comments and Suggestions for Authors

Dear authors, your study is original and interesting for the development of science. I believe that your article can be published, but you need to work on improving the text.

First, the Abstract does not contain the information required to publish articles in this journal. You need to completely redesign the Abstract.

Second, the article does not present the Methodology or Literature Review section (only briefly in the Introduction). The concept of Estonian development has a wide range of sources and research in various fields of science. I believe that in geographical sciences, the Concept of Sustainable Development also has a long history of study. It would be desirable to present its generalized characteristic.

Thirdly, the study you presented is detailed and detailed. But on the basis of only a cartographic method, drawing conclusions on sustainable development is not entirely correct. It is advisable to consider other methods. Especially, economic. (statistical).

Fourth, you submit your article to an international magazine. But your research only matters for one region of your country. You probably need to explain in the Discussion or Introduction section what is the meaning of your research for the world concept of sustainable development. Put an emphasis on that. Give examples for other countries or regions of the world. There is a lack of comparative analysis of your study.

I believe that your article needs to be finalized.

Author Response

We would like to express our sincere gratitude to you for your important research comments, which helped to enrich and improve the content of our work. Their value lies in the fact that they helped us deepen our understanding of the subject of study, identify additional aspects of the problem, and also suggest additional methods for data analysis. This made our research more comprehensive and informative. We wish you all the best and creative success.

Reviewer’s comment:

  1. Dear authors, your study is original and interesting for the development of science. I believe that your article can be published, but you need to work on improving the text. First, the Abstract does not contain the information required to publish articles in this journal. You need to completely redesign the Abstract.

Response to reviewer:

The abstract has been corrected:

Assessing the sustainable development of the tourism and recreational potential of rural areas of the Aktobe oblast of the Republic of Kazakhstan requires a comprehensive analysis of various aspects, including environmental, socio-economic and cultural sustainability. It is necessary to evaluate existing resources for tourism development, such as natural attractions, cultural heritage, infra-structure and transport accessibility. Analysis of sustainable tourism development includes envi-ronmental impact assessment, measures to conserve natural resources, as well as socio-cultural aspects, including the preservation of cultural heritage. When conducting a study on the sustainable development of rural tourism in the Aktobe oblast, statistical analysis and ranking were used. Key factors influencing the development of rural tourism in the region were identified using these methods. To assess the sustainable development of the tourism potential of rural areas, a balance method was used to assess the interaction of various aspects, such as environmental, economic and sociocultural sustainability and ensure a balance between them. Maps and charts were created for data analysis and visualization using ArcGIS 10. The results of the study allow us to identify priority areas for improving infrastructure. This study has practical significance for developing strategies and programs for the development of tourism in rural areas of the Aktobe oblast taking into account the principles of sustainable development.

Reviewer’s comment:

  1. Second, the article does not present the Methodology or Literature Review section (only briefly in the Introduction). The concept of Estonian development has a wide range of sources and research in various fields of science. I believe that in geographical sciences, the Concept of Sustainable Development also has a long history of study. It would be desirable to present its generalized characteristic

Response to reviewer:

The literary review has been added and expanded. But we have not found any literature about the development concepts of Estonia.

Thanks to the ecological and tourism potential, which is part of the natural resource potential, the territories fulfill their socio-cultural load, which contributes to their sustainable development. The need to study the problem of formation, development and assessment of the ecological and tourism potential of the region is determined not only by the need to diversify the use of resource opportunities, but to a greater extent by the turbulent state of the social and labor sphere. In addition, the lack of general methodological approaches to assessing the ecological and tourism potential does not contribute to its effective use and, as a consequence, to the stable socio-economic development of territories. The study of the natural prerequisites for the development of the tourism sector is traditionally the first stage in assessing the territory, since natural resources are one of the factors that predetermine its use.

The countryside, with its attractive appearance and the opportunity for city resi-dents to enjoy nature and a healthy lifestyle, satisfies their psychological needs and becomes a valuable resource in modern society. The commodification of this space, in-fluenced by consumer culture, increases its value as landscape, environment and products. However, this process can also lead to the abstraction and symbolization of rural space. Problems associated with excessive commodification are particularly felt in rural areas near large cities, reflecting a shift from overt to hidden forms of loss of rural identity .

Challenges relate to the unavailability of resources such as quality labor and in-vestment, as well as the inability to use local resources to develop rural tourism. The lack of planning and government support demonstrates insufficient attention to the resource approach in research related to rural tourism .

More in-depth research can be undertaken using a resource-based approach to understand the dynamic capabilities of rural destinations and identify how internal and external resources can be effectively identified, mobilized, harnessed and main-tained. This will maximize the benefits of rural tourism.

Modern changes in the preferences of consumers of recreational services, as well as the growth of environmental consciousness of vacationers, create demand for new forms of tourism products that use alternative recreational resources, including the resources of rural areas. In Kazakhstan, where agriculture is a diversified production, there is potential for the development of rural tourism as an additional factor in stimulating tourism in general. The culture, traditions and hospitality of the Kazakh people can give a new impetus to the development of tourism in rural areas, in accordance with the principles of sustainable development.

Aktobe oblast is one of the regions of Kazakhstan where rural tourism can become a priority sector of the economy. However, the territory of the region is characterized by insufficient recreational and geographical knowledge, which is one of the main factors hindering the development of the tourism industry, which is at the initial stage of formation. In this regard, the need arose for a comprehensive assessment of the environmental and tourism potential of the Aktobe region and determination of its regional specifics and prospects for use.

Reviewer’s comment:

  1. Thirdly, the study you presented is detailed and detailed. But on the basis of only a cartographic method, drawing conclusions on sustainable development is not entirely correct. It is advisable to consider other methods. Especially, economic. (statistical).

Response to reviewer:

When writing the article, not only the cartographic method was used to visualize data on maps, but also balance and ranking methods were used to analyze and evaluate the sustainable development of tourism in rural areas. The balance method made it possible to evaluate various aspects of the tourism potential of rural areas, taking into account both their advantages and limitations. The ranking method was used to set priorities and identify key factors affecting the sustainable development of rural tourism. Thus, the combination of these methods made it possible to obtain a deeper and more comprehensive analysis of the state and potential of tourism development in the rural areas under consideration.

Reviewer’s comment:

  1. Fourth, you submit your article to an international magazine. But your research only matters for one region of your country. You probably need to explain in the Discussion or Introduction section what is the meaning of your research for the world concept of sustainable development. Put an emphasis on that. Give examples for other countries or regions of the world. There is a lack of comparative analysis of your study.

Response to reviewer:

Corrected according to the comments:

Rural tourism is part of a variety of tourism types such as ecotourism, farm tour-ism, adventure tourism, gastronomic tourism and cultural tourism, forming a complex and multi-faceted sector with an ever-expanding variety of opportunities. Rural tour-ism covers three key aspects: the material characteristics of the countryside, the inter-action of tourists with these characteristics, and the cultural or symbolic value of the area. It is a form of tourism located in rural areas that brings together a variety of ac-tivities and services to develop and revitalize areas.

The development of rural tourism in each country depends on its unique natural and resource potential, as well as on the historical heritage, cultural attractions, gas-tronomic features and natural attractions. Rural tourism is currently expanding rap-idly in various countries around the world. In addition to well-known destinations such as Italy, Germany, France, China, Poland and the Baltic countries, best practices and experiences in rural tourism are being actively explored. Rural tourism has simi-larities with other types of tourism, and many of them can be successfully implement-ed in rural areas. Rural tourism can provide a variety of activities and experiences similar to spiritual, cultural and adventure tourism, including forms such as farm tourism. However, it is important to consider that too many attractions can dilute the uniqueness of the experience. The process of redefining rural tourism at the national level, as it has been done, for example, in Malaysia and Portugal, shows that develop-ing a new definition in a specific context can help better convey the experience of rural tourism destinations. This allows to identify the unique characteristics of each destina-tion and emphasize them as part of tourism promotion. As a result, although achieving a global consensus on the definition of rural tourism may be difficult, a definition spe-cific to each country or region may be more realistic and useful

Reviewer’s comment:

I believe that your article needs to be finalized.

Article corrected according to the comments

Reviewer 3 Report

Comments and Suggestions for Authors

The subject is very interesting, but the paper lacks coherence and structure.

First, there should be an overview of the state of rural tourism in Kazakhstan, emphasizing current policies for tourism development, for a better understanding of the context.

Materials and methods - this section is quite long, with a lot of information and methods mentioned, but in the end I did not quite understood what method was used for what exactly.

Rural tourism is based to a large extend on folklore, traditions and crafts of the rural population, which the current paper does not consider when evaluating the potential for development. The analysis should clearly indicate what are the areas with a high concentration of tourism resources but lacking tourism facilities and infrastructure. 

Please see further comments on the manuscript. 

Comments on the Quality of English Language

Moderate editing of the English language is required. 

Author Response

We would like to express our sincere gratitude to you for your important research comments, which helped to enrich and improve the content of our work. Their value lies in the fact that they helped us deepen our understanding of the subject of study, identify additional aspects of the problem, and also suggest additional methods for data analysis. This made our research more comprehensive and informative. We wish you all the best and creative success.

Reviewer’s comment:

  1. The subject is very interesting, but the paper lacks coherence and structure. First, there should be an overview of the state of rural tourism in Kazakhstan, emphasizing current policies for tourism development, for a better understanding of the context.

Response to reviewer:

Corrected in accordance with the comments:

Rural tourism in Kazakhstan is a promising direction that attracts the attention of both public and private structures within the framework of tourism development policy. Here is an overview of the current state of rural tourism in Kazakhstan, taking into account the current tourism development policy.

In recent years, a number of programs for the development of rural tourism have been launched in Kazakhstan, such as “Baitak Zher”, “On your own through Semirechye”, etc. These programs are aimed at creating new tourist routes, developing hotel and restaurant businesses, training local residents in the field of tourism and other activities. The “Baytak Zher” project, launched in the fall of 2022, is being successfully implemented in three districts of the Akmola oblast: Zerenda, Sandyktau and Korgalzhyn. Under the leadership of this project, 20 entrepreneurs completed a training incubation program covering the basics of eco- and ethnotourism, financial and project planning, as well as methods of promoting their own brand.

Although rural tourism is a promising industry, there are serious problems:

  1. Insufficient infrastructure. Rural areas often lack sufficient tourism infrastructure such as hotels, road networks, public transport and other amenities making it difficult to attract tourists;
  2. Lack of funding. Investments in the development of rural tourism are limited due to a lack of financial resources, both from the state and from private investors;

3.Low awareness and marketing. Many rural areas face a lack of awareness of the potential of rural tourism and lack effective marketing strategies to attract tourists;

  1. Lack of qualified personnel. In rural areas there are not enough qualified tourism specialists, which makes it difficult to provide quality services and manage tourism enterprises;

5.Problems of preserving the natural environment. Intensive tourism has a negative impact on the natural environment of rural areas if appropriate measures are not taken to protect the environment and sustainable use of resources;

  1. Sociocultural aspects. Some rural communities will face challenges in adapting to the arrival of tourists, including changing lifestyles, preserving traditions and maintaining privacy.

Solving these problems requires joint efforts on the part of the state, local authorities, entrepreneurs and communities in order to develop and implement rural tourism development strategies that take into account the needs of all stakeholders and ensure sustainable development.

Reviewer’s comment:

  1. Materials and methods - this section is quite long, with a lot of information and methods mentioned, but in the end I did not quite understood what method was used for what exactly.

Response to reviewer:

During the article, the cartographic method was used to visualize data on maps, balance and ranking methods for analyzing and evaluating the sustainable development of tourism in rural areas. The balance method made it possible to evaluate various aspects of the tourism potential of rural areas, taking into account both their advantages and limitations. The ranking method was used to set priorities and identify key factors affecting the sustainable development of rural tourism. Thus, the combination of these methods made it possible to obtain a deeper and more comprehensive analysis of the state and potential of tourism development in the rural areas under consideration.

Reviewer’s comment:

Rural tourism is based to a large extend on folklore, traditions and crafts of the rural population, which the current paper does not consider when evaluating the potential for development. The analysis should clearly indicate what are the areas with a high concentration of tourism resources but lacking tourism facilities and infrastructure. 

Response to reviewer:

We will not be able to fully disclose ethnotourism, but for the development of rural tourism we have considered the elements of Jailoo tourism:

The development of rural tourism is closely related to gastronomic characteristics and types of agricultural services of the rural population. In the regions of the Aktobe oblast, peasant farms are engaged in the production of meat and milk as the main activity. However, they are not included in the process of providing tourism and excursion services to the population everywhere, and only 20% are engaged in searching for an additional source of income through attracting tourists. In this regard, the development of rural tourism in the regions can be associated with Jailoo tourism. Jailoo is a place convenient for summer cattle keeping. Jailoo tourism includes a special synthesis of rural and ethnotourism, which is typical for all regions of Kazakhstan, including the Aktobe region. The characteristic lowland landscapes are considered unique compared to grasslands in other countries such as Kyrgyzstan. An attractive feature of Jailoo tourism is the healing grace in the summer and the traditional flavor of nomadic life: fresh air, spring water, organic food from meat and dairy products. The natural attractions of Zhailau tourism are harmoniously combined with the historical and cultural heritage, creating a unique aura of immersion in the original culture of nomads: an overnight stay in a yurt, folk songs, national games and rituals, horseback riding, master classes on folk crafts and national handicrafts made of wool and leather and other local materials. Tourists of Jailoo tourism are proposed to be classified according to their affiliation with the main purpose of the tour as follows:

  1. Tourists whose main purpose is kumiss treatment (Koumiss treatment is the general name of the healing process that uses fresh mare’s milk (saumal) and koumiss) and shubat treatment (Shubat (camel milk) can be recommended to people suffering from many diseases);
  2. Transit tourists stopping overnight on their way to other regions of the country or neighboring countries;
  3. Tourists relaxing on the river banks and briefly landing on Zhailau for an excursion.

For the sustainable development of Jailoo tourism, it is necessary to constantly cultivate summer pastures by all interested parties in the country: to maintain them in an attractive condition for tourists, both in natural and social terms.

Reviewer’s comment:

Please see further comments on the manuscript. Object of study. This section offers little valuable information. Natural, social and economic background is missing.

Response to reviewer:

Corrected in accordance with the comments:

Aktobe oblast is located in the western part of Kazakhstan. The length of the territory from west to east is about 800 km, from the north to the south about 700 km. Area 300,629 km². Population 924,845 people (2022) [45]. The Aktobe oblast occupies an intermediate position between the Caspian lowland in the west, the Ustyurt plateau in the south, the Turan lowland in the southeast and the southern slopes of the Ural Mountains in the north. Its territory is mainly represented by plains intersected by river valleys that rise to a height of 100 to 200 meters. Mugodzhary extends in the central part of the region. In the west of the region there is the Podural Plateau, which in the southwest turns into the Caspian Lowland. In the southeast of the region there are massifs of hilly sands - the Aral Karakum and Big and Small Badgers. In the northeast of the region stretches the Turgai plateau, riddled with ravines. Characterizing the climate of the Aktobe oblast, it should be noted that it has a pronounced continental character. Winters here are cold, and summers are hot and dry. Dry winds and dust storms often occur during the summer, and snowstorms are common in winter. The average temperature in July is around +22.5°C in the north-west and up to +25°C in the south-east, while in January it drops to -16°C and -25.5°C respectively. Precipitation varies between about 300 mm in the northwest and from 125 to 200 mm in the center and south of the region. The growing season here ranges from 175 to 190 days a year. The rivers of the Aktobe oblast, belonging to the endorheic basins of the Caspian Sea and small lakes, originate in Mugodzhary.

Aktobe oblast consists of 12 districts. Aktobe oblast is a region with a developed industrial and agricultural base, where agriculture is actively progressing. In parallel with this, measures are being taken to stimulate the development of the tourism sector, including the sector of rural tourism.

Reviewer’s comment:

this map doesn't provide valuable information. It lacks legend. Since you mention in the text the name of some districts, they should also be included on this map

Response to reviewer:

The map has been changed according to the remark.

Reviewer’s comment:

The international reader may not be familiar with the classes of water quality in Kazakhstan. Therefore, a brief explanation on what class 3 and 4 means is necessary.

Response to reviewer:

Corrected in accordance with the comments:

The unified system for classifying water quality in water bodies is divided into six water use classes with a gradual transition from class 1 of “best quality” water to class 6 of “worst quality.” Each water use class is characterized by its own water use category depending on the formed ecological potential of the water body. For water use for the purpose of recreation and recreation, water classes 1,2,3 are recommended.

Reviewer’s comment:

According to the legend, the only accommodation facilities are the camping sites. does this mean there are no hotels/ boarding houses etc? Or what exactly do you mean by public catering?

Response to reviewer:

Due to comments, the figure has been changed:

We meant the locations in general (hotels, campgrounds, etc.), public catering – means public catering establishments

Reviewer 4 Report

Comments and Suggestions for Authors

The article submitted for review is written in clear and understandable language. The authors attempted to assess the sustainable development of tourism and recreation potential of rural areas in the studied area.

The authors analyzed the elements of the tourism potential of this area that may be important for this development. They provided a clear description of the indicators used and the indicators that were used in this study.

All these aspects are described in the article and may become an inspiration for other researchers who could use them in similar activities.

The only thing missing for me in this article is an analysis of whether similar studies have already been carried out in this aspect and what their results were, whether there are descriptions of these results in the literature on the subject.

Author Response

We would like to express our sincere gratitude to you for your important research comments, which helped to enrich and improve the content of our work. Their value lies in the fact that they helped us deepen our understanding of the subject of study, identify additional aspects of the problem, and also suggest additional methods for data analysis. This made our research more comprehensive and informative. We wish you all the best and creative success.

Reviewer’s comment:

The article submitted for review is written in clear and understandable language. The authors attempted to assess the sustainable development of tourism and recreation potential of rural areas in the studied area.

The authors analyzed the elements of the tourism potential of this area that may be important for this development. They provided a clear description of the indicators used and the indicators that were used in this study.

All these aspects are described in the article and may become an inspiration for other researchers who could use them in similar activities.

The only thing missing for me in this article is an analysis of whether similar studies have already been carried out in this aspect and what their results were, whether there are descriptions of these results in the literature on the subject.

Response to reviewer:

The literature review section has been added according to your comments.

Round 2

Reviewer 2 Report

Comments and Suggestions for Authors

Dear authors, you have seriously improved your article. I believe that in this form your work may be published in a journal.

The drawings have become brighter and more understandable. I understand that you are engaged in a certain field of science. I think that your point of view has the right to publish. Although I do not support your attitude to the tourism sector, as a specialist in the economics of tourism.

Reviewer 3 Report

Comments and Suggestions for Authors

Thank you for considering some of the comments I made.

Comments on the Quality of English Language

The use of the English language must be improved.